# Distinguishing non severe cases of dengue from COVID-19 in the context of co-epidemics: A cohort study in a SARS-CoV-2 testing center on Reunion island

**Antoine Joubert**[1ʘ], **Fanny Andry**[1,2,3ʘ], **Antoine Bertolotti**[1,2,4], **Frédéric Accot**[1], **Yatrika Koumar**[1,2,3], **Florian Legrand**[1,3], **Patrice Poubeau**[1,2,3], **Rodolphe Manaquin**[1,2,3], **Patrick Gérardin**[4ʘ]*, **Cécile Levin**[1,2,3ʘ]*

**1** COVID-19 testing center, Centre Hospitalier Universitaire de la Réunion, Saint Pierre, Reunion, France, **2** Department of Infectious Diseases and Tropical Medicine, Centre Hospitalier Universitaire de la Réunion, Saint Pierre, Reunion, France, **3** City to Hospital Outpatient Clinic for the care of COVID-19, Centre Hospitalier Universitaire de la Réunion, Saint Pierre, Reunion, France, **4** Center for Clinical Investigation–Clinical Epidemiology (CIC 1410), Institut National de la Santé et de la Recherche Médicale (INSERM), Centre Hospitalier Universitaire de la Réunion, Saint Pierre, Reunion, France

ʘ These authors contributed equally to this work.
\* patrick.gerardin@chu-reunion.fr (PG); levincecile@gmail.com (CL)

**Data Availability Statement:** All relevant data are within the manuscript and its Supporting Information files.

## Abstract

### Background

As coronavirus 2019 (COVID-19) is spreading globally, several countries are handling dengue epidemics. As both infections are deemed to share similarities at presentation, it would be useful to distinguish COVID-19 from dengue in the context of co-epidemics. Hence, we performed a retrospective cohort study to identify predictors of both infections.

### Methodology/Principal findings

All the subjects suspected of COVID-19 between March 23 and May 10, 2020, were screened for COVID-19 within the testing center of the University hospital of Saint-Pierre, Reunion island. The screening consisted in a questionnaire surveyed in face-to-face, a nasopharyngeal swab specimen for the Severe Acute Respiratory Syndrome Coronavirus-2 (SARS-CoV-2) reverse transcription polymerase chain-reaction and a rapid diagnostic orientation test for dengue. Factors independently associated with COVID-19 or with dengue were sought using multinomial logistic regression models, taking other febrile illnesses (OFIs) as controls. Adjusted Odds ratios (OR) and 95% Confidence Intervals (95%CI) were assessed. Over a two-month study period, we diagnosed 80 COVID-19, 61 non-severe dengue and 872 OFIs cases eligible to multivariate analysis. Among these, we identified delayed presentation (>3 days) since symptom onset (Odds ratio 1.91, 95% confidence interval 1.07–3.39), contact with a COVID-19 positive case (OR 3.81, 95%CI 2.21–6.55) and anosmia (OR 7.80, 95%CI 4.20–14.49) as independent predictors of COVID-19, body ache (OR 6.17, 95%CI 2.69–14.14), headache (OR 5.03, 95%CI 1.88–13.44) and retro-orbital pain (OR 5.55, 95%CI 2.51–12.28) as independent predictors of dengue, while

**Funding:** The authors received no specific funding for this work.

**Competing interests:** The authors have declared that no competing interests exist.

smoking was less likely observed with COVID-19 (OR 0.27, 95%CI 0.09–0.79) and upper respiratory tract infection symptoms were associated with OFIs.

## Conclusions/Significance

Although prone to potential biases, these data suggest that non-severe dengue may be more symptomatic than COVID-19 in a co-epidemic setting with higher dengue attack rates. At clinical presentation, nine basic clinical and epidemiological indicators may help to distinguish COVID-19 or dengue from each other and other febrile illnesses.

## Author summary

As coronavirus 2019 (COVID-19) is spreading globally, several countries are facing dengue epidemics with the fear the two plagues might overburden their healthcare systems. On Reunion island, southwestern Indian ocean: dengue virus is circulating since 2004 under an endemo-epidemic pattern with yearly outbreaks peaking between March and May since 2015, whereas Severe Acute Respiratory Syndrome Coronavirus-2 (SARS-CoV-2), the pathogen responsible of COVID-19, emerged in March 2020, imported from the Bahamas. COVID-19 and dengue are deemed two clinically similar entities, especially within the first two days from symptom onset. In this context, we conducted a cohort study between March 23 and May 10, 2020, within a SARS-CoV-2 testing center, aimed at identifying the factors discriminating both infections. Surprisingly, we found that non-severe dengue was more symptomatic than mild to moderate COVID-19. Indeed, we found body ache, headache and retro-orbital pain to be indicative of dengue, whereas contact with a COVID-19 positive case, anosmia, delayed presentation (>3 days post symptom onset) and absence of active smoking were indicative of COVID-19. These findings highlight the need for accurate diagnostic tools and not to jeopardize dengue control in areas wherever COVID-19 dengue co-epidemics have the potential to wrought havoc to the healthcare system.

## Introduction

During the past decades, there have been growing concerns about the risks of overlapping epidemics and co-infections with emergent viruses, especially with arboviruses that can share the same *Aedes* mosquito vector [1,2]. Yet, surprisingly, since the 2009 flu pandemic, the differential diagnosis between influenza and dengue has been scarcely investigated [3].

As Severe Acute Respiratory Syndrome Coronavirus-2 (SARS-CoV-2) is spreading globally, several countries are handling dengue epidemics, with fear for their healthcare systems and most vulnerable populations [4]. Thus, to differentiate between the two diagnoses may be challenging and lead to misdiagnosis, which may occasion both delays in treatment and preventable deaths, but also inadequate isolation measures with the potential to trigger outbreaks, especially in the healthcare setting [4].

On Reunion island, a French overseas department located in the Indian ocean, best known to have hosted one of the largest chikungunya outbreaks and harbor a highly comorbid population [5,6], dengue virus (DENV) is circulating since 2004 under an endemo-epidemic pattern with outbreaks usually peaking between March and May, these intensifying with yearly

upsurges since 2015 [7]. In 2020, the first cases of coronavirus 2019 (COVID-19) were detected on the island by March 11, six days before the French authorities decreed the lockdown.

In this context, a new case of COVID-19 and dengue co-infection was reported [8]. Anticipating that the differential diagnosis between the two infections would be challenging, we designated a retrospective cohort study aimed at identifying the clinical and epidemiological profiles of SARS-CoV-2 and DENV infections to guide their management and mitigate the impact of COVID-19 pandemic surge on the island.

## Methods

### Ethics statement

Outpatients presenting consecutively at the SARS-CoV-2 testing center were informed of the study orally and by means of an information sheet. Adult people, like the children under 18 years (with the additional verbal consent of their parent or legal guardian) who expressed no opposition, were asked to answer a questionnaire and surveyed in face-to-face by a nurse, in accordance to the French legislation on bioethics for retrospective researches. Patient's medical records were retrospectively reviewed, and de-identified data were collected in standardized forms according to the MR-004 procedure of the *Commission Nationale de l'Informatique et des Libertés* (the French information protection commission). The ethical character of this study on previously collected data was approved by the Scientific Committee for COVID-19 research of the CHU Réunion and de-identified data were registered on the Health Data Hub.

### Study design, setting and population

We conducted a retrospective cohort study using prospectively collected data between March 23 and May 10, 2020, on all subjects screened for the COVID-19 within the UDACS (*Unité de Dépistage Ambulatoire du COVID-19 Sud*) of Saint-Pierre, one of the two SARS-CoV-2 testing centers of the *Centre Hospitalier Universitaire* (CHU) *Réunion*. When SARS-CoV-2 emerged on the island, the dengue epidemic was already burgeoning, the UDACS was placed in the second line of the reception system for COVID-19 patients, the frontline being the emergency units and the dedicated hospital for COVID-19 patients, the CHU Félix Guyon, located in Saint-Denis, whereby are the prefecture and the international airport. People without symptoms were excluded from the study.

### Data collection

The items of the questionnaire included information on demographics, occupation, risk factors, comorbidities, intra-household and individual exposure to SARS-CoV-2, individual symptoms and treatment. Temperature, pulse rate, respiratory rate and peripheral oxygen saturation ($SpO_2$) were measured upon the consultation, as well as clinical symptoms, including verification of the presence of cough and anxiety.

### Diagnostic procedures

All the attendees were screened by a skilled nurse for SARS-CoV-2 using a nasopharyngeal swab inserted and held in one nostril until reaching the posterior wall of the nasopharynx for about twenty seconds [9]. The sample was processed for a SARS-CoV-2 reverse transcription-polymerase chain reaction (RT-PCR) using the Allplex 2019-nCov assay (Seegene, Seoul, Republic of Korea) or an in-house kit (CNR Pasteur), targeting N, RdRP and E genes, or N and IP2/IP4 targets of RdRP, respectively. In addition, each patient suspected of dengue was tested for NS1 antigen using an OnSite Duo dengue Ag-IgG-IgM rapid diagnostic test (CTK

Biotech, San Diego, CA, USA) and if negative further explored with a DENV RT-PCR or a dengue serology depending on the date of symptom onset.

## Statistical analysis

Given the research purpose, COVID-19-dengue co-infections at clinical presentation were excluded from the analysis. Other febrile illnesses (OFIs) were defined as patients tested negative for SARS-CoV-2 and further considered as unrelated to dengue, either clinically, virologically, or serologically. COVID-19, dengue and OFI subjects were compared using Chi square or Fisher exact tests, as appropriate. Univariable and multivariable multinomial logistic regression models were fitted within Stata14 (StataCorp, College Station, Texas, USA) to identify both the independent predictors of COVID-19 and dengue taking OFIs as controls.

Crude and adjusted odds ratios (OR) and 95% confidence intervals (95%CI) were assessed using the binomial and Cornfield methods, respectively.

For all these analyses, observations with missing data were ruled out and a *P*-value less than 0.05 considered statistically significant.

The full details of the methods can be found in the S1 text file. The results were reported following the STROBE guideline (S1 STROBE checklist).

## Results

Between March 23 and May 10, 2020, 1,715 subjects presented at the UDACS for screening or diagnosis purposes. Of these, 370 incoming patients were screened opportunistically for COVID-19 as part of an expanded screening week targeting admissions to our hospital (75% asymptomatic, all tested negative), and 332 were fully asymptomatic subjects (44% with the notion of a COVID-19 contact, of whom 6 tested positive; 53% healthcare workers, of whom 2 tested positive; 5 tested positive without notion of COVID-19 contact nor an occupational exposure). Both these populations were excluded from the study, leaving 1,013 outpatients eligible to the analysis. The study population is shown in Fig 1.

The characteristics of the 1,013 symptomatic subjects eligible to analysis are presented in Table 1.

The hospitalization rates (at least one night) for the COVID-19 and dengue patients were higher than those observed for the patients affected by OFIs (17.5% and 8.2%, respectively *versus* 1.5%, $P < 0.001$). Among 32 patients that were hospitalized, 2 COVID-19 patients out of 14 met the criteria for COVID-19 pneumonia and 5 dengue patients out of 5 had dengue warning signs but none had severe dengue at clinical presentation. No COVID-19 dengue co-infection was observed at clinical presentation.

COVID-19 patients presented later in their evolution compared to the subjects affected by dengue or OFIs (time elapsed since symptom onset, 7.5 days *versus* 4.2 days or 6.3 days, $P < 0.001$). The average levels of temperature, pulse rate, respiratory rate and $SpO_2$ did not differ between the three groups of patients.

Univariable analysis proposed contact with a COVID-19+ case, recent return from travel abroad ($< 15$ d), fever, ageusia, anosmia (loss of smell) and delayed presentation ($> 3$ d) since symptom onset as candidate predictors for COVID-19, active smoking as candidate protective factor against COVID-19 (S1 Table). Previous episode of dengue, fever, body ache (*i.e.*, muscle pain, backache with tightness/stiffness), ageusia, gut symptoms (*i.e.*, nausea, vomiting, dyspepsia, eructation or abdominal pain), metallic taste, fatigue, headache and retro-orbital pain were identified as candidate predictors for dengue whereas recent return from travel abroad and cough, as candidate protective factors against dengue. Interestingly, upper respiratory tract infection (URTI) symptoms (*i.e.*, sore throat, runny nose, nasal congestion or sneezing) were

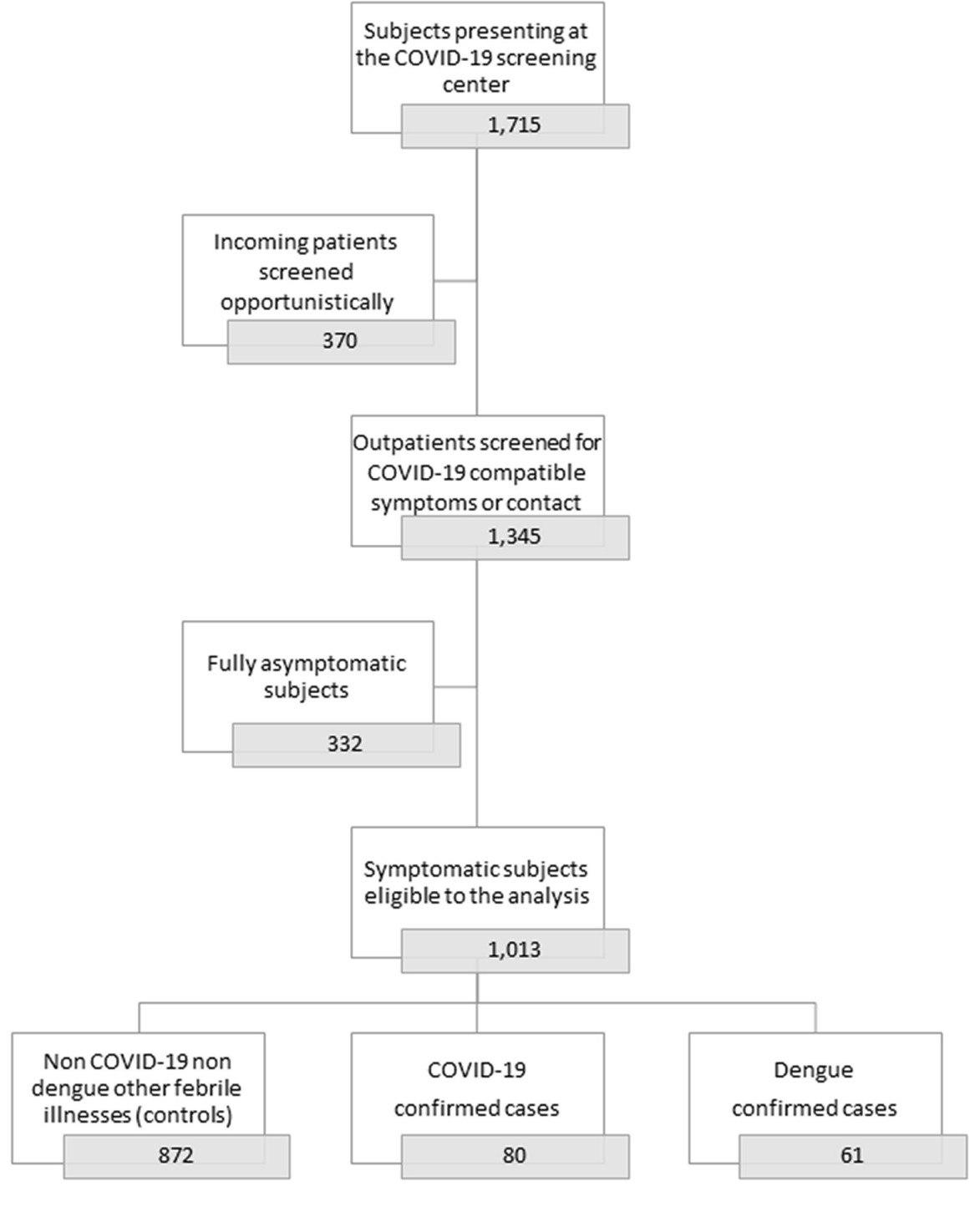

**Fig 1. Study population.** Flow chart of the study population.

identified as candidate protective factors against both diagnoses, which made these rather pre-dictors of OFIs (S1 Table).

Multivariable analysis identified delayed presentation (>3 d) since symptoms onset, contact with a COVID-19 positive case and anosmia as independent predictors of COVID-19, body

**Table 1. Characteristics of 1,013 subjects consulting a COVID-19 screening center during the COVID-19 dengue co-epidemics, Reunion island, Saint-Pierre, March 23-May 10, 2020.**

| Outcomes | Other febrile illnesses (n = 872) | | COVID-19 (n = 80) | | Dengue (n = 61) | | |
|---|---|---|---|---|---|---|---|
| Variables | N | (%) | N | (%) | N | (%) | P value |
| Male gender | 343 | 39.3 | 33 | 41.2 | 31 | 50.8 | 0.205 |
| Age (years), μ ± sd | 38.7 | 16.2 | 39.2 | 18.4 | 42.0 | 13.4 | 0.280 |
| 0–30 (Q1) | 253 | 29.0 | 28 | 35.0 | 8 | 13.1 | 0.002 |
| 31–41 (Q2) | 247 | 28.3 | 10 | 12.5 | 26 | 42.6 | |
| 42–54 (Q3) | 221 | 25.3 | 27 | 33.7 | 13 | 21.3 | |
| 55–94 (Q4) | 151 | 17.3 | 15 | 18.7 | 14 | 22.9 | |
| Contact with a COVID-19 positive case | 231 | 26.5 | 42 | 52.5 | 6 | 9.8 | < 0.001 |
| Return from travel abroad < 15 days | 202 | 23.2 | 42 | 53.2 | 6 | 9.8 | < 0.001 |
| Previous dengue episode | 32 | 3.7 | 6 | 7.6 | 9 | 14.7 | 0.001 |
| Comorbidities$^\S$ | 435 | 49.9 | 34 | 42.5 | 31 | 50.8 | 0.437 |
| Morbid obesity (body mass index $\geq$ 40 kg/m$^2$) | 19 | 2.2 | 0 | 0.0 | 2 | 3.3 | 0.359 |
| Active smoking[†] | 146 | 16.8 | 4 | 5.2 | 12 | 19.7 | 0.022 |
| Fever | 374 | 42.9 | 45 | 56.2 | 59 | 96.7 | < 0.001 |
| Duration of fever (days), μ ± sd | 3.34 | 3.15 | 3.43 | 3.35 | 3.03 | 2.88 | 0.799 |
| Cough | 435 | 49.9 | 36 | 45.0 | 17 | 27.9 | 0.003 |
| Duration of cough (days), μ ± sd | 5.44 | 5.71 | 2.14 | 12.84 | 5.79 | 7.98 | 0.099 |
| Dyspnea/Shortness of breath | 204 | 23.4 | 13 | 16.3 | 13 | 21.3 | 0.332 |
| Duration of dyspnea (days), μ ± sd | 4.14 | 4.50 | 5.44 | 8.43 | 7.75 | 5.25 | 0.376 |
| Body ache[‡] | 339 | 38.9 | 32 | 40.0 | 52 | 85.2[‡] | < 0.001 |
| Duration of pain (days), μ ± sd | 3.89 | 3.69 | 4.34 | 5.49 | 2.90 | 2.72 | 0.088 |
| Diarrhea | 179 | 20.2 | 19 | 23.7 | 13 | 21.3 | 0.746 |
| Duration of liquid stools (days), μ ± sd | 2.70 | 2.62 | 4.50 | 3.79 | 2.25 | 3.14 | 0.099 |
| Gut symptoms[¶] | 44 | 5.0 | 4 | 5.0 | 13 | 21.3 | < 0.001 |
| Ageusia | 84 | 9.6 | 25 | 31.2 | 11 | 18.0 | < 0.001 |
| Duration of ageusia (days), μ ± sd | 3.93 | 4.16 | 4.73 | 3.32 | 3.25 | 2.01 | 0.263 |
| Metallic taste (dysgeusia) | 4 | 0.5 | 0 | 0.0 | 2 | 3.3 | 0.068 |
| Anosmia | 67 | 7.7 | 28 | 35.0 | 3 | 4.9 | < 0.001 |
| Duration of anosmia (days), μ ± sd | 4.35 | 4.45 | 4.22 | 3.59 | 1.00 | 1.00 | 0.199 |
| Fatigue | 370 | 42.4 | 38 | 47.5 | 49 | 80.3 | < 0.001 |
| Duration of fatigue (days), μ ± sd | 4.29 | 4.20 | 6.48 | 5.75 | 3.44 | 3.00 | 0.027 |
| Headache | 410 | 47.1 | 31 | 38.7 | 56 | 91.8 | < 0.001 |
| Duration of headache (days), μ ± sd | 3.95 | 3.94 | 4.69 | 5.61 | 3.02 | 2.74 | 0.324 |
| Retro-orbital pain | 26 | 3.0 | 1 | 1.2 | 17 | 27.9 | < 0.001 |
| URTI symptoms[#] | 459 | 52.6 | 31 | 38.7 | 20 | 32.8 | 0.001 |
| Duration of rhinorrhea (days), μ ± sd | 4.45 | 4.47 | 5.33 | 3.69 | 2.10 | 0.91 | 0.036 |
| Duration of sore throat (days), μ ± sd | 4.17 | 3.98 | 4.00 | 3.27 | 6.20 | 8.22 | 0.995 |
| Presentation > 3 days after symptom onset | 481 | 57.2 | 54 | 70.1 | 24 | 40.0 | 0.002 |
| Time elapsed since symptom onset (days), μ ± sd | 6.27 | 6.25 | 7.54 | 6.50 | 4.18 | 4.57 | < 0.001 |
| Need for physical examination at presentation | 131 | 15.0 | 8 | 10.1 | 19 | 31.1 | 0.001 |
| Dry cough upon testing | 10 | 1.1 | 2 | 2.6 | 1 | 1.6 | 0.315 |
| Anxiety upon testing | 17 | 1.9 | 0 | 0.0 | 0 | 0.0 | 0.459 |
| Frontal temperature (˚C), μ ± sd | 37.11 | 0.92 | 36.98 | 0.99 | 37.33 | 1.27 | 0.337 |
| Cardiac rate (pulses per minute), μ ± sd | 86.84 | 16.46 | 86.38 | 16.80 | 89.89 | 18.60 | 0.520 |
| Respiratory rate (cycles per minute), μ ± sd | 17.56 | 4.88 | 17.39 | 5.69 | 18.09 | 4.98 | 0.479 |
| SpO$_2$ (%), μ ± sd | 97.85 | 1.08 | 97.23 | 1.47 | 97.72 | 1.11 | 0.002 |

*(Continued)*

**Table 1.** (*Continued*)

| Outcomes | Other febrile illnesses (n = 872) | | COVID-19 (n = 80) | | Dengue (n = 61) | | |
|---|---|---|---|---|---|---|---|
| Variables | N | (%) | N | (%) | N | (%) | *P* value |
| Hospitalization | 13 | 1.5 | 14 | 17.5 | 5 | 8.2 | < 0.001 |
| Length of Stay (days), μ ± sd | 1.4 | 0.7 | 9.9 | 7.1 | 1.0 | 0.7 | < 0.001 |

Data are numbers, column percentages, and *P* values for Chi2 or Fisher's exact tests, unless specified as means, standard deviations, and *P* values for Kruskal-Wallis tests.

$ 15: Urgent Medical Aid Service (SAMU).

§ diabetes, hypertension, cardiovascular disease, chronic obstructive pulmonary disease, asthma, or cancer.

† Current smoker, as compared to never smoker and past smoker.

‡ muscle pain or backache with tightness and/or stiffness.

¶ nausea, vomiting, dyspepsia, eructation or abdominal pain.

# sore throat, runny nose, nasal congestion, or sneezing.

ache, headache and retro-orbital pain as independent predictors of dengue, while active smoking was less likely observed with COVID-19 and URTI symptoms were indicative of OFIs (Table 2).

Further analyses weighted on the inverse probability of hospitalization were inconsistent to confirm the robustness of the protective association of active smoking with COVID-19 (S2 and S3 Tables).

A sensitivity analysis restricted to the patients with COVID-19 or with dengue confirmed anosmia, URTI symptoms and delayed presentation (>3 d) on the one hand, body ache, fatigue, headache, retro-orbital pain and rapid presentation (≤ 3 d) on the other hand, as discriminating factors between the two infections (S4 Table).

Data supporting the analyses are available online (S1 Data). Supportive statistical metadata are provided in a.txt supplemental file (S2 Data).

**Table 2. Independent predictors in multivariate analysis distinguishing COVID-19 and dengue from other febrile illnesses among 972 subjects consulting a COVID-19 screening center during the COVID-19 dengue co-epidemics, Reunion island, Saint-Pierre, March 23-May 10, 2020.**

| Outcomes (versus other febrile illnesses as controls*) | COVID-19 (n = 74) | | | | | Dengue (n = 60) | | | | |
|---|---|---|---|---|---|---|---|---|---|---|
| Predictors | n | CIR, % | aOR | 95% CI | *P* value | n | CIR, % | aOR | 95% CI | *P* value |
| Contact with a COVID-19 positive case | 40 | 15.33 | 3.81 | 2.21–6.55 | < 0.001 | 6 | 2.30 | 0.81 | 0.31–2.09 | 0.663 |
| Active smoking† | 4 | 2.53 | 0.27 | 0.09–0.79 | 0.017 | 12 | 7.59 | 1.39 | 0.65–2.94 | 0.391 |
| Cough | 32 | 6.82 | 0.82 | 0.47–1.42 | 0.474 | 17 | 3.62 | 0.38 | 0.19–0.73 | 0.003 |
| Body ache‡ | 29 | 7.09 | 1.12 | 0.66–2.14 | 0.564 | 52 | 12.71 | 6.17 | 2.69–14.14 | < 0.001 |
| Anosmia | 26 | 27.96 | 7.80 | 4.20–14.49 | < 0.001 | 3 | 3.23 | 0.47 | 0.12–1.75 | 0.258 |
| Headache | 28 | 5.69 | 0.79 | 0.45–1.38 | 0.403 | 55 | 11.18 | 5.03 | 1.88–13.44 | 0.001 |
| Retro-orbital pain | 1 | 2.27 | 0.45 | 0.05–3.74 | 0.458 | 17 | 38.64 | 5.55 | 2.51–12.28 | < 0.001 |
| URTI symptoms# | 28 | 5.63 | 0.52 | 0.30–0.91 | 0.021 | 20 | 4.02 | 0.49 | 0.26–0.93 | 0.028 |
| Presentation > 3 days after symptom onset | 54 | 9.69 | 1.91 | 1.07–3.39 | 0.027 | 24 | 4.31 | 0.74 | 0.40–1.36 | 0.339 |

Multinomial logistic regression model with other non COVID-19 non dengue febrile illnesses* taken as controls. Data are numbers, cumulative incidence rates (CIR) expressed as percentages, adjusted odd ratios (aOR), 95% confidence intervals (95% CI) and *P* values for Wald tests.

† Current smokers, as compared to never smokers and past smokers.

‡ muscle pain or backache with tightness and/or stiffness.

# sore throat, runny nose, nasal congestion, or sneezing.

The indicators of performance of the model are as follows: Bayesian information criterion -5733, Goodness of fit chi-2 test's probability 0.823, areas under the receiver operating characteristic curves 0.783 and 0.877, respectively.

## Discussion

COVID-19 and dengue are two clinically similar entities, especially within the first 24 to 48 hours from symptom onset [10]. In a context of co-epidemics, our cohort study, conducted within a SARS-CoV-2 testing center upon mild to moderate cases of COVID-19 and non-severe cases of dengue identified several key distinctive features for both infections. Among the clinically discriminant variables at presentation, retro-orbital pain, body ache and head-ache were strong predictors of dengue while anosmia was the only predictor of COVID-19 and URTI symptoms were indicative of OFIs. To a lesser extent, gut symptoms other than diarrhea, dysgeusia and fatigue were suggestive of dengue whereas cough referred to another diagnosis (OFIs or COVID-19), albeit found in nearly a third of dengue. Among the epidemio-logical variables, the contact with a COVID-19+ case and a delayed presentation beyond three days of symptom onset were predictive of COVID-19, a rapid presentation within three days was suggestive of dengue, while active smoking was less likely observed with COVID-19 or associated with OFIs. These elements are summarized in the S1 Fig.

Our findings reveal several unexpected differences at the presentation to hospital between COVID-19 or dengue as compared to OFIs, and between COVID-19 and dengue, dengue appearing at first glance more symptomatic and with a more abrupt onset than COVID-19 or OFIs in the setting of a SARS-CoV-2 testing center.

These discrepancies might reflect first a selection bias, the more symptomatic cases of both infec-tions having been referred primarily to the emergency units, these redirecting the COVID-19 cases towards the Saint-Denis referral hospital for quarantine purpose. This could be arguably deduced from weighting on the inverse probability of hospitalization, which was on average 2.5-fold higher than that observed from the UDACS, all through the study period. Doing so abrogated, for instances, the effects of a delayed presentation and the protection of active smoking for the predic-tion of COVID-19. Together with the fact that the dengue epidemic was more active in the south-ern part of the island, this fuels the idea that time to presentation in our study partly stemmed from differences in recruitment driven both by the organization and access to care. Importantly, weight-ing the analysis also strengthened the odds ratios of a contact with a COVID-19+ case for the same, as well as those of headache for the prediction of dengue. These elements suggest that this putative selection bias was more pronounced on epidemiological than on clinical variables.

Second, our results might also be affected by a misclassification bias, which may arise both from the poor sensitivity of SARS-CoV-2 RT-PCR and rapid NS1 antigen, rather than from the false positive rates.

Third, given the fear of COVID-19 at that time, we cannot rule out the possibility of a reporting bias, as some patients may have declared URTI symptoms or cough in excess just to be tested for SARS-CoV-2. Consistent with this, are the relatively high percentages of cough and URTI symptoms among dengue cases, as well as the totality of anxiety cases upon testing observed within the OFI group, for instances.

Together with the abovementioned sources of bias, a lack of power might have reduced the capability to shed light on other discriminating factors. However, we believe that this study faithfully reflects the real epidemiological situation on Reunion island at that time, given diag-nostic practices and means that were commonly used in this era of uncertainty, which is unlikely to have biased the overall sense of our results.

These being said, our findings are also in agreement with the literature.

First, the fact that dengue was more symptomatic than COVID-19 fulfills both the concept of "force of infection" and the trade-off model according to which, the time spent in the sus-ceptible group to an infectious disease is inversely correlated to its incidence [11]. Under this model, the virulence (i.e., ability to cause illness, lethality) grows with the transmission rate until

it reaches a plateau [12]. Consistent with these assumptions, according to *Santé Publique France* reports, the attack rate observed over the study period was 22-fold higher for dengue (≈905 per 100.000 inhabitants) than for COVID-19 (≈41 per 100.000 inhabitants). This was explained by the recent introduction of DENV-1 serotype (March 2019) complicating five years of DENV-2 circulation [7], cases of secondary dengue, the effectiveness of the lockdown to slow the progression of COVID-19 and the fact that SARS-CoV-2 impacted at that time mainly "healthy" individuals (travelers and their relatives). In this framework, the relevance of body ache, headache and retro-orbital pain at presentation for the differential diagnosis between COVID-19 and dengue accounts for the involvement of dengue in the general and digestive spheres, as proposed by Nacher et *al.* in a recent opinion paper, COVID-19 being more pronounced in the respiratory sphere [10]. Interestingly, we also found one COVID-19+ case who was tested negative for dengue suffering retro-orbital pain, as previously reported in Taiwan [13].

Second, our cohort study supports the high positive predictive values and specificities of the contact with a COVID-19+ case and anosmia for the diagnostic of COVID-19, which is congruent with risk prediction models developed for healthcare workers in Italy [14] and findings from the *Coranosmia* cohort study in France [15], respectively.

Together with the abovementioned putative selection bias, the delayed presentation to hospital of COVID-19 cases, as compared to dengue, might also illustrate the mild ("pauci-symptomatic") character of COVID-19 illness during the first pandemic surge on Reunion island, as well as some consecutive lags in contact tracing. Overall, the individuals who did not feel or only felt slightly sick with COVID-19 might not have felt the need to be tested. This hypothesis is supported by the fact that the cases of COVID presented later than the OFIs, despite theoretically similar symptoms and a proportion of asymptomatic two times lower.

Interestingly, active smoking was less likely to be observed with COVID-19 as compared to OFIs or dengue, but this apparent protective effect was not robust as suggested above. Moreover, it was not replicated for asymptomatic SARS-CoV-2 infections, nor was it shown to protect from contracting illness with COVID-19. This finding seems paradoxical given recent evidence shows that Angiotensin-converting enzyme 2 (ACE2), the SARS-CoV-2 entry receptor, is overexpressed in smoker's bronchial and alveolar epithelia, which should increase the risk of infection [16–18]. Whether this finding results from abovementioned misclassification or reporting bias deserves further studies. Notwithstanding, this fuels the smoker's paradox according to which active smokers were first underreported among the patients hospitalized for COVID-19 in several countries [19]. In line with this paradox, current smokers were less likely to be infected in a recent meta-analysis [20].

In conclusion, our cohort study identified several factors distinguishing non severe dengue from COVID-19 at clinical presentation in a context of recent dengue endemicity and first introduction of SARS-CoV-2. Although prone to potential biases, these data suggest that non severe dengue may be more symptomatic at presentation than COVID-19 in a co-epidemic setting with higher dengue attack rates, a pattern that might also result from different forces of infection (lesser exposure to SARS-CoV-2 than to DENV). Whether these findings may serve other regions facing co-epidemics, deserves more investigations, development, and validation of more accurate diagnostic tools. These findings highlight also the need not to jeopardize dengue control wherever COVID-19 dengue co-epidemics have the potential to wrought havoc to the healthcare system [21].

## Supporting information

**S1 STROBE checklist.**
(DOC)

**S1 Fig. Predictors associated with COVID-19, dengue, and other febrile illnesses.** Venn diagram summarizing the predictors for COVID-19, dengue and other febrile illnesses. Predictors for COVID-19 are displayed in the bottom left circle of the Venn diagram, predictors for dengue in the top circle, and predictors for non-COVID-19 non-dengue other febrile illnesses in the bottom right circle. Independent predictors are in bold characters, crude predictors that do not resist to multiple adjustments are in thin characters.
(TIF)

**S1 Table. Crude predictors in bivariate analysis distinguishing COVID-19 and dengue from other febrile illnesses among 1,013 subjects consulting a COVID-19 screening center during the COVID-19 dengue co-epidemics, Reunion island, Saint-Pierre, March 23-May 10, 2020.** * Other non COVID-19 non dengue febrile illnesses. Data are numbers, cumulative incidence rates (CIR) expressed as percentages, crude odd ratios (cOR), 95% confidence intervals (95% CI) and *P* values for Wald tests. † Current smokers, as compared to never smokers and past smokers ‡ muscle pain or backache with tightness and/or stiffness; ⁋ nausea, vomiting, dyspepsia, eructation or abdominal pain # sore throat, runny nose, nasal congestion, or sneezing. N.A: not assessed (incalculable).
(DOCX)

**S2 Table. Independent predictors in weighted multivariate analysis (scenario 1) distinguishing COVID-19 and dengue from other febrile illnesses among 972 subjects consulting a COVID-19 screening center during the COVID-19 dengue co-epidemics, Reunion island, Saint-Pierre, March 23-May 10, 2020.** Multinomial logistic regression model with other non COVID-19 non dengue febrile illnesses* taken as controls. In this model, the probability of OFIs cases to be hospitalized was set at 16% (speculated). Data are numbers, weighted cumulative incidence rates (wCIR) expressed as percentages, survey-adjusted odd ratios (s-aOR), 95% confidence intervals (95% CI) and *P* values for Wald tests. † Current smokers, as compared to never smokers and past smokers. ‡ muscle pain or backache with tightness and/or stiffness. # sore throat, runny nose, nasal congestion, or sneezing. The indicators of performance of the model are unavailable with the *svy* option in Stata.
(DOCX)

**S3 Table. Independent predictors in weighted multivariate analysis (scenario 2) distinguishing COVID-19 and dengue from other febrile illnesses among 972 subjects consulting a COVID-19 screening center during the COVID-19 dengue co-epidemics, Reunion island, Saint-Pierre, March 23-May 10, 2020.** Multinomial logistic regression model with other non COVID-19 non dengue febrile illnesses* taken as controls. In this model, the probability of OFIs cases to be hospitalized was set at 16% (speculated). Data are numbers, weighted cumulative incidence rates (wCIR) expressed as percentages, survey-adjusted odd ratios (s-aOR), 95% confidence intervals (95% CI) and *P* values for Wald tests. † Current smokers, as compared to never smokers and past smokers. ‡ muscle pain or backache with tightness and/or stiffness. # sore throat, runny nose, nasal congestion, or sneezing. The indicators of performance of the model are unavailable with the *svy* option in Stata.
(DOCX)

**S4 Table. Sensitivity analysis.** Crude predictors in bivariate analysis distinguishing COVID-19 from dengue from after exclusion of other febrile illnesses among 141 subjects consulting a COVID-19 screening center during the COVID-19 dengue co-epidemics, Reunion island, Saint-Pierre, March 23-May 10, 2020. * Other non COVID-19 non dengue febrile illnesses. Data are numbers, row percentages, and *P* values for Chi2 or Fisher's exact tests, unless specified as means, standard deviations, and *P* values for Mann-Whitney tests. † Current smokers,

as compared to never smokers and past smokers. [‡] muscle pain or backache with tightness and/or stiffness. [¶] nausea, vomiting, dyspepsia, eructation or abdominal pain. [#] sore throat, runny nose, nasal congestion, or sneezing.
(DOCX)

**S1 Data. Dataset.** This file includes the data supporting the analyses.
(XLSX)

**S2 Data. Supportive statistical metadata.** This file includes all supporting metadata that have been produced in reply to reviewer's comments to argue the findings.
(TXT)

**S1 Text. Methodological appendix.** Full detail of the methods.
(DOCX)

## Acknowledgments

The authors are indebted to the staffs of the department of Infectious Disease and Tropical Medicine and the SARS-CoV-2 testing center, especially the nurses who performed the survey. They thank the biologists of the CHU for timely diagnosis and the attendees for kind interest in research.

## Author Contributions

**Conceptualization:** Fanny Andry, Patrick Gérardin, Cécile Levin.

**Data curation:** Patrick Gérardin.

**Formal analysis:** Antoine Bertolotti, Patrick Gérardin.

**Investigation:** Antoine Joubert, Fanny Andry, Antoine Bertolotti, Frédéric Accot, Yatrika Koumar, Florian Legrand, Patrice Poubeau, Rodolphe Manaquin, Cécile Levin.

**Methodology:** Patrick Gérardin.

**Project administration:** Fanny Andry.

**Resources:** Patrice Poubeau.

**Software:** Patrick Gérardin.

**Supervision:** Patrick Gérardin.

**Validation:** Fanny Andry, Patrick Gérardin.

**Visualization:** Cécile Levin.

**Writing – original draft:** Antoine Joubert, Patrick Gérardin, Cécile Levin.

**Writing – review & editing:** Antoine Joubert, Fanny Andry, Antoine Bertolotti, Frédéric Accot, Yatrika Koumar, Florian Legrand, Patrice Poubeau, Rodolphe Manaquin, Patrick Gérardin, Cécile Levin.

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
