## [Decision Letter · Decision Letter 0]

17 Feb 2021

Dear Dr. Gérardin,

Thank you very much for submitting your manuscript "Distinguishing non severe cases of dengue from COVID-19 in the context of co-epidemics: a cohort study in a SARS-CoV-2 testing center on Reunion island" for consideration at PLOS Neglected Tropical Diseases. As with all papers reviewed by the journal, your manuscript was reviewed by members of the editorial board and by several independent reviewers. In light of the reviews (below this email), we would like to invite the resubmission of a significantly-revised version that takes into account the reviewers' comments. 

We cannot make any decision about publication until we have seen the revised manuscript and your response to the reviewers' comments. Your revised manuscript is also likely to be sent to reviewers for further evaluation.

Sincerely,

Johan Van Weyenbergh

Associate Editor

Victor S. Santos

Deputy Editor

Reviewer's Responses to Questions

**Key Review Criteria Required for Acceptance?**

**Methods**

-Are the objectives of the study clearly articulated with a clear testable hypothesis stated?

-Is the study design appropriate to address the stated objectives?

-Is the population clearly described and appropriate for the hypothesis being tested?

-Is the sample size sufficient to ensure adequate power to address the hypothesis being tested?

-Were correct statistical analysis used to support conclusions?

-Are there concerns about ethical or regulatory requirements being met?

Reviewer #1: -Are the objectives of the study clearly articulated with a clear testable hypothesis stated? Yes

-Is the study design appropriate to address the stated objectives? Yes

-Is the population clearly described and appropriate for the hypothesis being tested? Yes

-Is the sample size sufficient to ensure adequate power to address the hypothesis being tested? Yes

-Were correct statistical analysis used to support conclusions? Yes

-Are there concerns about ethical or regulatory requirements being met? Yes

Reviewer #2: small number recruited, and hence analyses are not reliable.

Reviewer #3: The rapid assay test used to diagnose dengue virus infection was the OnSiteTM Duo dengue Ag-IgG-IgM rapid diagnostic test based on the NS1 antigen. If test results were negative a DENV RT-PCR or a dengue serology were performed. Can the authors comment on the sensitivity and specificity of the point of care assay? This is critical to make sure dengue cases were not misclassified or over diagnosed through an assay that may carry a high false positive rate. 

 The methods section does not detail any characteristics of participants diagnosed with OFIs. Were other diagnoses entertained, for example chikungunya, or TB or leptospirosis or strep throat for example? There should be an explanation of how the control group was defined. Were these patients who had a febrile illness and tested negative for COVID-19 and dengue? Were OFIs diseases that were not associated with any specific diagnosis? 

 The methods section states that patients with co-infections were excluded. Can this statement be clarified? Does that mean participants who had either COVID-19 or dengue and some other condition? If so how was the other disease diagnosed? Is there an estimate of how many participants were not enrolled because of co-infections?

**Results**

-Does the analysis presented match the analysis plan?

-Are the results clearly and completely presented?

-Are the figures (Tables, Images) of sufficient quality for clarity?

Reviewer #1: -Does the analysis presented match the analysis plan? Yes

-Are the results clearly and completely presented? Yes

-Are the figures (Tables, Images) of sufficient quality for clarity? Yes

Reviewer #2: (No Response)

Reviewer #3: What diseases were representative of OFIs? There has been an overall decrease in circulation of influenza and other viral respiratory pathogens during the COVID-19 pandemic, so it is unusual that respiratory symptoms were predominantly associated with OFIs since there was an overall decline in the circulation of viral respiratory pathogens, particularly those that cause fever such as influenza. It is also unusual that respiratory symptoms were protective factors against a COVID-19 diagnosis since respiratory symptoms are heavily associated with COVID-19 (sore throat, runny nose, nasal congestion or sneezing cited in the manuscript). Is it possible that there were false negative results from RT-PCR for SARS CoV-2? Were the nasal swabs self collected? What was the expected sensitivity and specificity of the SARS CoV-2 RT PCR assay used? What do the authors propose as a potential explanation for these findings? It is also unusual that cough would be so prevalent in dengue patients. As alluded to in the discussion, misclassification of participants may have been considerable. 

 Why would smoking be associated with a lower likelihood of COVID-19? What would explain such an observation? It was thought in the early days of the pandemic that smoking would make pulmonary disease worse. One potential explanation is the small sample size of smokers, in table S1 there are only 4 smokers in the COVID-19 category and 12 in the dengue category. 

 COVID-19 symptoms vary by age. What was the age of study participants, and how did disease presentation vary by age? It appears that age distribution was not significant for COVID-19 cases but it was significant in dengue cases, with the younger age group being more protected against dengue. There is no comment about this finding in the results or discussion but it would be worth providing a potential explanation for this finding. 

 Cough was a protective factor against dengue in table 1, but not necessarily associated with COVID-19 which is unusual. Also unusual that URTI symptoms would be a protective risk factor for COVID-19, it was also protective for dengue but that would be expected. This is highly suggestive of potential misclassification, with potentially COVID-19 patients being transferred to the OFIs group because of a negative test result, which would explain the unusual high frequency of URTIs in the control group.

 The frequencies of the clinical/ demographic parameters are not reported for OFIs. They should be displayed in a table along with the other two illnesses.

**Conclusions**

-Are the conclusions supported by the data presented?

-Are the limitations of analysis clearly described?

-Do the authors discuss how these data can be helpful to advance our understanding of the topic under study?

-Is public health relevance addressed?

Reviewer #1: -Are the conclusions supported by the data presented? Yes

-Are the limitations of analysis clearly described? Yes

-Do the authors discuss how these data can be helpful to advance our understanding of the topic under study? Yes

-Is public health relevance addressed? Yes

Reviewer #2: Dengue versus COVID-19 at presentation are so different, and the conclusion in the abstract and manuscript is misleading. “these data suggest that non-severe 45dengue may be more symptomatic than COVID-19 in a co-epidemic setting with higher dengue attack 46rates. At clinical presentation, eightbasic clinical and epidemiological indicators may help to 47distinguish COVID-19 or dengue from each other and other febrile illnesses.”

Dengue presents in younger persons. COVID-19 affects the bulk of all age groups, but severe cases mainly only occur in older people. COVID-19 is usually mild in the first days and then can progress to severe disease usually around day 10 of illness with rapid progression to death if no access to oxygen and more supportive care is available. 

It would also be misleading to rely on clinical and epidemiological indicators. The key message should be that all cases should have a diagnostic work-up for COVID-19, as underdiagnosing COVID-19 could propagate further transmission. The consequences of unmitigated transmission is huge. Misdiagnosing dengue does not have so many repercussions as the vast majority of cases improve spontaneously, and there is no exponential growht in onward transmission.

Reviewer #3: A study limitation is the absence of an asymptomatic tested population. Since both dengue and COVID-19 have high asymptomatic infection rates, it would have been of interest to explore positivity rates for both conditions in this setting, as a research opportunity. It is understandable, however, that this population was not evaluated given that the main hypothesis focused on clinical parameters associated with each condition.

**Editorial and Data Presentation Modifications?**

Reviewer #1: Line 191: COVID+ should be corrected to COVID-19+

Line 246? OFis should be corrected to OFIs

Reviewer #2: 

Reviewer #3: More information about OFIs is needed in the manuscript.

The issue of potential misclassification of COVID-19 cases as OFIs should be further addressed, particularly when there are findings that are not compatible with the clinical conditions represented. 

The degree of fever could potentially differentiate between the two conditions, the analysis does not look at high versus low fever. Stratification of clinical findings by age would also be something of interest, particularly for COVID as the disease differs across age groups. Some comment about dengue prevalence in certain age groups is worth including, as younger age groups seemed less likely to have dengue.

**Summary and General Comments**

Reviewer #1: As Severe Acute Respiratory Syndrome Coronavirus-2 (SARS-CoV-2) is spreading globally, worldwide growing concerns about the risks of overlapping epidemics and co-infections with emergent viruses, especially with arboviruses, have being intensifying. Here, the authors designated a retrospective cohort study aimed at identifying the clinical and epidemiological profiles of SARS-CoV-2 and DENV infections to guide their management and mitigate the impact of COVID-19 pandemic surge locally. The paper is generally interesting to the scientific community and infectious disease experts in particular. A strength of this study is to prepare the healthcare system to better and rapid distinguish between both SARS-CoV-2 or DENV infections, avoiding delays in treatment and preventable deaths.

Reviewer #2: Dengue versus COVID-19 at presentation are so different, and the conclusion in the abstract and manuscript is misleading. “these data suggest that non-severe 45dengue may be more symptomatic than COVID-19 in a co-epidemic setting with higher dengue attack 46rates. At clinical presentation, eightbasic clinical and epidemiological indicators may help to 47distinguish COVID-19 or dengue from each other and other febrile illnesses.”

Dengue presents in younger persons. COVID-19 affects the bulk of all age groups, but severe cases mainly only occur in older people. COVID-19 is usually mild in the first days and then can progress to severe disease usually around day 10 of illness with rapid progression to death if no access to oxygen and more supportive care is available. 

It would also be misleading to rely on clinical and epidemiological indicators. The key message should be that all cases should have a diagnostic work-up for COVID-19, as underdiagnosing COVID-19 could propagate further transmission. The consequences of unmitigated transmission is huge. Misdiagnosing dengue does not have so many repercussions as the vast majority of cases improve spontaneously, and there is no exponential growht in onward transmission.

Reviewer #3: The manuscript proposes an interesting analysis of clinical relevance, which is to find demographic/ clinical parameters that may help differentiate between dengue infection and COVID-19 disease in a setting highly endemic for arboviral illnesses. A comparison group of other febrile illnesses is included in the analysis but this group needs to be better characterized, with definitions and more data provided about potential diagnoses in this control population. More data about clinical findings in the OFI population should also be provided in the tables. The authors should try to further address the issue of potential misclassification of diagnoses, since some of the findings do not appear to be biologically plausible, such as respiratory symptoms being protective against a COVID-19 diagnosis for example. This merits further investigation. Some of the other results are highly plausible. The degree of fever could potentially differentiate between the two conditions, the analysis does not look at high versus low fever. Stratification of clinical findings by age would be something of interest, particularly for COVID as the disease differs across age groups.
---

## [Editor Report · Decision Letter 1]

15 Mar 2021

Dear Dr. Gérardin,

We are pleased to inform you that your manuscript 'Distinguishing non severe cases of dengue from COVID-19 in the context of co-epidemics: a cohort study in a SARS-CoV-2 testing center on Reunion island' has been provisionally accepted for publication in PLOS Neglected Tropical Diseases.

Best regards,

Johan Van Weyenbergh

Associate Editor

Victor S. Santos

Deputy Editor

---

## [Editor Report · Acceptance letter]

22 Apr 2021

Dear Dr Gérardin,

We are delighted to inform you that your manuscript, "Distinguishing non severe cases of dengue from COVID-19 in the context of co-epidemics: a cohort study in a SARS-CoV-2 testing center on Reunion island," has been formally accepted for publication in PLOS Neglected Tropical Diseases.

Best regards,

Shaden Kamhawi

co-Editor-in-Chief

Paul Brindley

co-Editor-in-Chief
